# Beyond Separability: Analyzing the Linear Transferability of Contrastive Representations to Related Subpopulations

**Jeff Z. HaoChen**
Stanford University
jhaochen@stanford.edu

**Colin Wei**
Stanford University
colinwei@stanford.edu

**Ananya Kumar**
Stanford University
ananya@cs.stanford.edu

**Tengyu Ma**
Stanford University
tengyuma@stanford.edu

## Abstract

Contrastive learning is a highly effective method for learning representations from unlabeled data. Recent works show that contrastive representations can transfer across domains, leading to simple state-of-the-art algorithms for unsupervised domain adaptation. In particular, a linear classifier trained to separate the representations on the source domain can also predict classes on the target domain accurately, even though the representations of the two domains are far from each other. We refer to this phenomenon as *linear transferability*. This paper analyzes when and why contrastive representations exhibit linear transferability in a general unsupervised domain adaptation setting. We prove that linear transferability can occur when data from the same class in different domains (e.g., photo dogs and cartoon dogs) are more related with each other than data from different classes in different domains (e.g., photo dogs and cartoon cats) are. Our analyses are in a realistic regime where the source and target domains can have unbounded density ratios and be weakly related, and they have distant representations across domains.

## 1 Introduction

In recent years, contrastive learning and related ideas have been shown to be highly effective for representation learning [Chen et al., 2020a,b, He et al., 2020, Caron et al., 2020, Chen et al., 2020c, Gao et al., 2021, Su et al., 2021, Chen and He, 2020]. Contrastive learning trains representations on *unlabeled data* by encouraging positive pairs (e.g., augmentations of the same image) to have closer representations than negative pairs (e.g., augmentations of two random images). The learned representations are almost *linearly separable*: one can train a linear classifier on top of the fixed representations and achieve strong performance on many natural downstream tasks [Chen et al., 2020a]. Prior theoretical works analyze contrastive learning by proving that semantically similar datapoints (e.g., datapoints from the same class) are mapped to geometrically nearby representations [Arora et al., 2019, Tosh et al., 2020, 2021, HaoChen et al., 2021]. In other words, representations form clusters in the Euclidean space that respect the semantic similarity; therefore, they are linearly separable for downstream tasks where datapoints in the same semantic cluster have the same label.

Intriguingly, recent empirical works show that contrastive representations carry richer information *beyond* the cluster memberships—they can transfer across domains in a linear way as elaborated below. Contrastive learning is used in many unsupervised domain adaptation algorithms[Thota

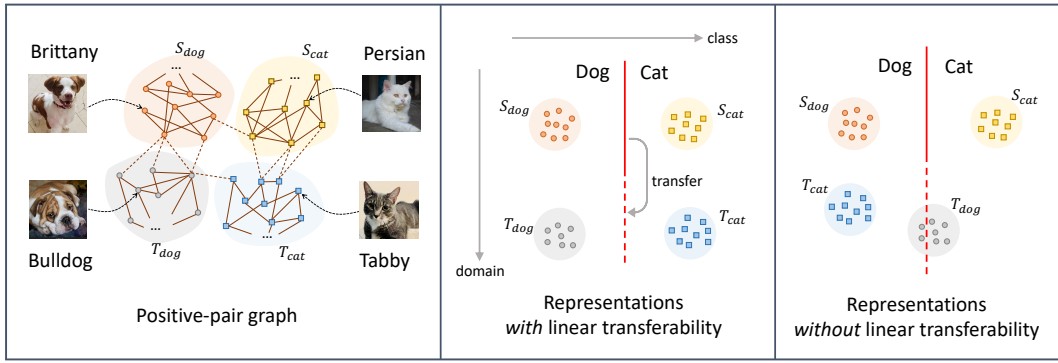

Figure 1: **The linear transferability of representations.** We demonstrate the linear transferability of representations when the unlabeled data contains images of two breeds of dogs (Brittanys, Bulldogs) and two breeds of cats (Persians, Tabbies). **Left:** A visualization of the positive-pair graph with four semantic clusters. Inter-cluster edges (dashed) have a much smaller weight than intra-cluster edges (solid). Inter-cluster edges between two breeds of dogs (or cats) have more weight than that between a dog cluster and a cat cluster. **Middle and right:** A visualization of two different types of representations: both have linear separability, but only the middle one has linear transferability. The red line is the decision boundary of a dog-vs-cat linear classifier trained in the representation space on *labeled* Brittanys ($S_{\text{dog}}$) vs. Persians ($S_{\text{cat}}$) images. The representation has linear transferability if this classifier is accurate on *unlabeled* Bulldogs ($T_{\text{dog}}$) vs. Tabbies ($T_{\text{cat}}$) images.

and Leontidis, 2021, Sagawa et al., 2022] and the transferability leads to simple state-of-the-art algorithms [Shen et al., 2022, Park et al., 2020, Wang et al., 2021]. In particular, Shen et al. [2022] observe that the relationship between two clusters can be captured by their relative positions in the representation space. For instance, as shown in Figure 1 (middle), suppose $S_{\text{dog}}$ and $S_{\text{cat}}$ are two classes in a *source* domain (e.g., Brittany dogs and Persian cats), and $T_{\text{dog}}$ and $T_{\text{cat}}$ are two classes in a *target* domain (e.g., Bulldogs and Tabby cats). A *linear* classifier trained to separate the representations of $S_{\text{dog}}$ and $S_{\text{cat}}$ turns out to classify $T_{\text{dog}}$ and $T_{\text{cat}}$ as well. This suggests the four clusters of representations are not located in the Euclidean space randomly (e.g., as in Figure 1 (right)), but rather in a more aligned position as in Figure 1 (middle). We refer to this phenomenon as the *linear transferability* of contrastive representations.

This paper analyzes when and why contrastive representations exhibit linear transferability in a general unsupervised domain adaptation setting. Evidently, linear transferability can only occur when clusters corresponding to the same class in two domains (e.g., Brittany dogs and Bulldogs) are somewhat *related* with each other. Somewhat surprisingly, we found that a weak relationship suffices: linear transferability occurs as long as corresponding classes in different domains are more related than different classes in different domains. Concretely, under this assumption (Assumptions 3.1 or 3.3), a linear head learned with labeled data on one domain (Algorithm 1) can successfully predict the classes on the other domain (Theorems 3.2 and 3.4). Notably, our analysis provably shows that representations from contrastive learning do not only encode cluster identities but also capture the inter-cluster relationship, hence explains the empirical success of contrastive learning for domain adaptation.

Compared to previous theoretical works on unsupervised domain adaptation [Shimodaira, 2000, Huang et al., 2006, Sugiyama et al., 2007, Gretton et al., 2008, Ben-David et al., 2010, Mansour et al., 2009, Kumar et al., 2020, Chen et al., 2020d, Cai et al., 2021], our results analyze a modern, practical algorithm with weaker and more realistic assumptions. We do not require bounded density ratios or overlap between the source and target domains, which were assumed in some classical works [Sugiyama et al., 2007, Ben-David et al., 2010, Zhang et al., 2019, Zhao et al., 2019]. Another line of prior works [Kumar et al., 2020, Chen et al., 2020d] assume that data is Gaussian or near-Gaussian, whereas our result allows more general data distribution. Cai et al. [2021] analyze pseudolabeling algorithms for unsupervised domain adaptation, but require that the same-class cross-domain data are more related with each other (i.e., more likely to form positive pairs) than cross-class same-domain data are. We analyze a contrastive learning algorithm with strong empirical

performance, and only require that the same-class cross-domain data are more related with each other than cross-class *cross-domain* data, which is intuitively and empirically more realistic as shown in Shen et al. [2022]. (See related work and discussion below Assumption 3.1 for details).

Technically, we significantly extend the framework of HaoChen et al. [2021] to allow distribution shift—our setting only has labels on one subpopulation of the data (the source domain). Studying transferability to unlabeled subpopulations requires both novel assumptions (Assumptions 3.1 and 3.3) and novel analysis techniques (as discussed in Section 4).

Our analysis also introduces a variant of the linear probe—instead of training the linear head with the logistic loss, we learn it by directly computing the average representations within a class, multiplied by a preconditioner matrix (Algorithm 1). We empirically test this linear classifier on benchmark datasets and show that it achieves superior domain adaptation performance in Section 5.

**Additional Related Works.** A number of papers have analyzed the linear separability of representations from contrastive learning [Arora et al., 2019, Tosh et al., 2020, 2021, HaoChen et al., 2021] and self-supervised learning [Lee et al., 2020], whereas we analyze the linear transferability. Shen et al. [2022] also analyze the linear transferability but only for toy examples where the data is generated by a stochastic block model. Their technique requires a strong symmetry of the positive-pair graph (which likely does not hold in practice) so that top eigenvectors can be analytically derived. Our analysis is much more general and does not rely on explicit, clean form of the eigenvectors (which is impossible for general graphs).

Empirically, pre-training on a larger unlabeled dataset and then fine-tuning on a smaller labeled dataset is one of the most successful approaches for handling distribution shift [Blitzer et al., 2007, Ziser and Reichart, 2018, 2017, Ben-David et al., 2020, Chen et al., 2012, Xie et al., 2020, Jean et al., 2016, Hendrycks et al., 2020, Kim et al., 2022, Kumar et al., 2022, Sagawa et al., 2022, Thota and Leontidis, 2021, Shen et al., 2022]. Recent advances in the scale of unlabeled data, such as in BERT and CLIP, have increased the importance of this approach [Wortsman et al., 2022, 2021]. Despite the empirical progress, there has been limited theoretical understanding of why pre-training helps domain shift. Our work provides the first analysis that shows pre-trained representations with a supervised linear head trained on one domain can provably generalize to another domain.

## 2 Preliminaries

In this section, we introduce the contrastive loss, define the positive-pair graph, and introduce the basic assumptions on the clustering structure in the positive-pair graph.

**Positive pairs.** Contrastive learning algorithms rely on the notion of "positive pairs", which are pairs of semantically similar/related data. Let $\mathcal{X}$ be the set of population data and $P_+$ be the distribution of positive pairs of data satisfying $P_+(x, x') = P_+(x', x)$ for any $x, x' \in \mathcal{X}$. We note that though a positive pair typically consists of semantically related data, the vast majority of semantically related pairs are *not* positive pairs. In the context of computer vision problems [Chen et al., 2020a], these pairs are usually generated via data augmentation on the same image.

For the ease of exposition, we assume $\mathcal{X}$ is a finite but large set (e.g., all real vectors in $\mathbb{R}^d$ with bounded precision) of size $N$. We use $P_{\mathcal{X}}$ to denote the marginal distribution of $P_+$, i.e., $P_{\mathcal{X}}(x) := \sum_{x' \in \mathcal{X}} P_+(x, x')$. Following the terminology in the literature [Arora et al., 2019], we call $(x, x')$ a "negative pair" if $x$ and $x'$ are independent random samples from $P_{\mathcal{X}}$.

**Generalized spectral contrastive loss.** Contrastive learning trains a representation function (feature extractor) by minimizing a certain form of contrastive loss. Formally, let $f : \mathcal{X} \to \mathbb{R}^k$ be a mapping from data to $k$-dimensional features. In this paper, we consider a more general version of the spectral contrastive loss proposed in HaoChen et al. [2021]. Let $I_{k \times k}$ be the $k$-dimensional identity matrix. We consider the following loss with regularization strength $\sigma > 0$:

$$\mathcal{L}_\sigma(f) = \mathop{\mathbb{E}}_{(x, x^+) \sim P_+} \left[ \left\| f(x) - f(x^+) \right\|_2^2 \right] + \sigma \cdot R(f), \tag{1}$$

where the regularizer is defined as $R(f) = \left\| \mathop{\mathbb{E}}_{x \sim P_{\mathcal{X}}} \left[ f(x) f(x)^\top \right] - I_{k \times k} \right\|_F^2$. The loss $\mathcal{L}_\sigma$ intuitively minimizes the closeness of positive pairs via its first term, while regularizing the representations'

covariance to be identity, avoiding all the representations to collapse to the same point. Simple algebra shows that $\mathcal{L}_\sigma$ recovers the original spectral contrastive loss when $\sigma = 1$ (see Proposition B.1 for a formal derivation). We note that this loss is similar in spirit to the recently proposed Barlow Twins loss [Zbontar et al., 2021].

**The positive-pair graph.**  One useful way to think of positive pairs is through a graph defined by their distribution. Let the *positive-pair graph* be a weighted undirected graph $G(\mathcal{X}, w)$ such that the vertex set is $\mathcal{X}$, and for $x, x' \in \mathcal{X}$, the undirected edge $(x, x')$ has weight $w(x, x') = P_+(x, x')$. This graph was introduced by [HaoChen et al. [2021]] as the augmentation graph when the positive pairs are generated from data augmentation. We introduce a new name to indicate the more general applications of the graph into other use cases of contrastive learning (e.g. see [Gao et al. [2021]]). We use $w(x) = P_\mathcal{X}(x) = \sum_{x' \in \mathcal{X}} w(x, x')$ to denote the total weight of edges connected to a vertex $x$. We call $\bar{A} \in \mathbb{R}^{N \times N}$ the *normalized adjacency matrix* of $G(\mathcal{X}, w)$ if $\bar{A}_{xx'} = w(x, x')/\sqrt{w(x)w(x')}$,[1] and call $\mathcal{L} := I_{N \times N} - \bar{A}$ the *Laplacian* of $G(\mathcal{X}, w)$.

## 2.1 Clustering assumptions

Previous work accredits the success of contrastive learning to the clustering structure of the positive-pair graph—because the positive pairs connect data with similar semantic contents, the graph can be partitioned into many semantically meaningful clusters. To formally describe the clustering structure of the graph, we will use the notion of expansion. For any subset $A$ of vertices, let $w(A) \triangleq \sum_{x \in A} w(x)$ be the total weights of vertices in $A$. For any subsets $A, B$ of vertices, let $w(A, B) \triangleq \sum_{x \in A, x' \in B} w(x, x')$ be the total weights between set $A$ and $B$. We abuse notation and use $w(x, B)$ to refer to $w(\{x\}, B)$ when the first set is a singleton.

**Definition 2.1** (Expansion). *Let $A, B$ be two disjoint subsets of $\mathcal{X}$. We use $\phi(A, B)$, $\bar{\phi}(A, B)$ and $\underline{\phi}(A, B)$ to denote the expansion, max-expansion and min-expansion from $A$ to $B$ respectively, defined as*

$$\phi(A, B) = \frac{w(A, B)}{w(A)}, \qquad \bar{\phi}(A, B) = \max_{x \in A} \frac{w(x, B)}{w(x)}, \qquad \underline{\phi}(A, B) = \min_{x \in A} \frac{w(x, B)}{w(x)} \tag{2}$$

*Note that $\underline{\phi}(A, B) \leq \phi(A, B) \leq \bar{\phi}(A, B)$.*

Intuitively, $\phi(A, B)$ is the average proportion of edges adjacent to vertices in $A$ that go to $B$, whereas the max-(min-)expansion is an upper (lower) bound of this proportion for each $x \in A$.

Our basic assumption on the positive-pair graph is that the vertex set $\mathcal{X}$ can be partitioned into $m$ groups $C_1, \ldots, C_m$ with small connections (expansions) across each other.

**Assumption 2.2** (Cross-cluster connections). *For some $\alpha \in (0, 1)$, we assume that the vertices of the positive-pair graph $G$ can be partition into $m$ disjoint clusters $C_1, \ldots, C_m$ such that for any $i \in [m]$,*

$$\bar{\phi}(C_i, \mathcal{X} \backslash C_i) \leq \alpha \tag{3}$$

We will mostly work with the regime where $\alpha \ll 1$. Intuitively, each $C_i$ corresponds to all the data with a certain semantic meaning or a class of interest. For instance, $C_i$ may contain dogs from a certain breed. Our assumption is slightly stronger than in [HaoChen et al. [2021]]. In particular, they assume that the average expansions cross clusters is small, i.e., $\sum_{i \in [m]} \phi(C_i, \mathcal{X} \backslash C_i) \cdot w(C_i) \leq \alpha$, whereas we assume that the max-expansion is smaller than $\alpha$ for each cluster. In fact, since $\sum_{i \in [m]} w(C_i) = 1$ and $\phi(C_i, \mathcal{X} \backslash C_i) \leq \bar{\phi}(C_i, \mathcal{X} \backslash C_i)$, Assumption 2.2 directly implies their assumption. However, we note that Assumption 2.2 is still realistic in many domains. For instance, any bulldog $x$ has way more neighbors that are still bulldogs than neighbors that are Brittany dog, which suggests the max-expansion between bulldogs and Brittany dogs is small.

We also introduce the following assumption about intra-cluster expansion that guarantees each cluster can not broken into two well-separated sub-clusters.

**Assumption 2.3** (Intra-cluster conductance). *For all $i \in [m]$, assume the conductance of the subgraph restricted to $C_i$ is large, that is, every subset $A$ of $C_i$ with at most half the size of $C_i$ expands to the rest:*

$$\forall A \subset C_i \text{ satisfying } w(A) \leq w(C_i)/2, \ \phi(A, C_i \backslash A) \geq \gamma. \tag{4}$$

---

[1] We index $\bar{A}$ by $(x, x') \in \mathcal{X} \times \mathcal{X}$. Generally, we will index the $N$-dimensional axis of an array by $x \in \mathcal{X}$.

We have $\gamma < 1$ and we typically work with the regime where $\gamma$ is decently large (e.g., $\Omega(1)$, or inverse polynomial in dimension[2] and much larger than the cross-cluster connections $\alpha$. This is the same regime where prior work HaoChen et al. [2021] guarantees the representations of clusters are linearly separable.

We also remark that all the assumptions are on the population positive-pair graph, which is sparse but has reasonable connected components (as partially evaluated in Wei et al. [2020]). The rest of the paper assumes access to population data, but the main results can be extended to polynomial sample results by levering a model class for representation functions with bounded Rademacher complexity as shown in HaoChen et al. [2021].[3]

## 3  Main Results on Linear Transferability

In this section, we analyze the *linear transferability* of contrastive representations by showing that representations encode information about the relative strength of relationships between clusters.

Let $S$ and $T$ be two disjoint subsets of $\mathcal{X}$, each formed by $r$ clusters corresponding to $r$ classes. We say a representation function has linear transferability from the *source domain* $S$ to the *target domain* $T$ if a linear head trained on labeled data from $S$ can accurately predict the class labels on $T$. E.g., the representations in Fig. 1 (middle) has linear transferability because the max-margin linear classifier trained on $S_{\text{dog}}$ vs. $S_{\text{cat}}$ also works well on $T_{\text{dog}}$ vs. $T_{cat}$. We note that linear separability is a different, weaker notion, which only requires the four groups of representations to be linearly separable from each other.

Mathematically, we assume that the source domain and target domain are formed by $r$ clusters among $C_1, \ldots, C_m$ for $r \leq m/2$. Without loss of generality, assume that the source domain consists of cluster $S_1 = C_1, \ldots, S_r = C_r$ and the target domain consists of $T_1 = C_{r+1}, \ldots, T_r = C_{2r}$. Thus, $S = \cup_{i \in [r]} S_i$ and $T = \cup_{i \in [r]} T_i$. We assume that the correct label for data in $S_i$ and $T_i$ is the cluster identity $i$. Contrastive representations are trained on (samples of) the entire population data (which includes all $C_i$'s). The linear head is trained on the source with labels, and tested on the target.

Our key assumption is that the source and target classes are related correspondingly in the sense that there are more same-class cross-domain connections (between $S_i$ and $T_i$) than cross-class cross-domain connections (between $S_i$ and $T_j$ with $i \neq j$), formalized below.

**Assumption 3.1** (Relative expansion). *Let $\rho \triangleq \min_{i \in [r]} \phi(T_i, S_i)$ be the minimum min-expansions from $T_i$ to $S_i$. For some sufficiently large universal constant $c$ (e.g., $c = 8$ works), we assume that $\rho \geq c \cdot \alpha^2$ and that*

$$\rho = \min_{i \in [r]} \underline{\phi}(T_i, S_i) \geq c \cdot \max_{i \neq j} \cdot \bar{\phi}(T_i, S_j) \tag{5}$$

Intuitively, equation (5) says that every vertex in $T_i$ has more edges connected to $S_i$ than to $S_j$. The condition $\rho \gtrsim \alpha^2$ says that the min-expansion $\rho$ is bigger than the square of max-expansion $\alpha$. This is reasonable because $\alpha \ll 1$ and thus $\alpha^2 \ll \alpha$, and we consider the min-expansion $\rho$ and max-expansion $\alpha$ to be somewhat comparable. In Section 3.1 we will relax this assumption and study the case when the average expansion $\phi(T_i, S_i)$ is larger than $\phi(T_i, S_j)$.

Our assumption is weaker than that in the prior work [Cai et al., 2021] which also assumes expansion from $S_i$ to $T_i$ (though their goal is to study label propagation rather than contrastive learning). They assume the same-class cross-domain conductance $\phi(T_i, S_i)$ to be larger than the *cross-class* same-domain conductance $\phi(S_i, S_j)$. Such an assumption limits the application to situations where the domains are far away from each other (such as DomainNet [Peng et al., 2019]).

Moreover, consider an interesting scenario with four clusters: photo dog, photo cat, sketch dog, and sketch cat. Shen et al. [2022] empirically showed that transferability can occur in the following two settings: (a) we view photo and sketch as domains: the source domain is photo dog vs photo cat, and the target domain is sketch dog vs sketch cat; (b) we view cat and dog as domains, whereas photo and

---

[2]E.g., suppose each cluster's distribution is a Gaussian distribution with covariance $I$, and the data augmentation is Gaussian blurring with a covariance $\frac{1}{d} \cdot I$, then the intra-cluster expansion is $\Omega(1)$ by Gaussian isoperimetric inequality [Bobkov et al., 1997]. The same also holds with a Lipschitz transformation of Gaussian.

[3]In contrast, the positive-graph built only on empirical examples will barely have any edges, and does not exhibit any nice properties. However, the sample complexity bound does not utilize the empirical graph at all.

sketch are classes: the source domain is photo dog vs sketch dog, and the target is photo cat vs sketch cat. The condition that cross-domain expansion is larger than cross-class expansion will fail to explain the transferability for one of these settings—if $\phi(\text{photo dog, sketch dog}) < \phi(\text{photo dog, photo cat})$, then it cannot explain (a), whereas if $\phi(\text{photo dog, sketch dog}) > \phi(\text{photo dog, photo cat})$, it cannot explain (b). In contrast, our assumption only requires conditions such as $\phi(\text{photo dog, sketch dog}) > \phi(\text{photo dog, sketch cat})$, hence works for both settings.

We will propose a simple and novel linear head that enables linear transferability. Let $\mathcal{P}_S$ be the data distribution restricted to the source domain.[4] For $i \in [r]$, we construct the following average representation for class $i$ in the source:[5]

$$b_i = \mathop{\mathbb{E}}_{x \sim \mathcal{P}_S} \left[ \mathbb{1}\left[ x \in S_i \right] \cdot f(x) \right] \in \mathbb{R}^k. \tag{6}$$

One of the most natural linear head is to use the average feature $b_i$'s as the weight vector for class $i$, as in many practical few shot learning algorithms [Snell et al., 2017].[6] That is, we predict

$$g(x) = \arg\max_{i \in [r]} \langle f(x), b_i \rangle. \tag{7}$$

This classifier can transfer to the target under relatively strong assumptions (see the special cases in the proof sketch in Section 4), but is vulnerable to complex asymmetric structures in the graph. To strengthen the result, we consider a variant of this classifier with a proper preconditioning.

To do so, we first define the representation covariance matrix which will play an important role:

$$\Sigma = \mathop{\mathbb{E}}_{x \sim P_{\mathcal{X}}} \left[ f(x) f(x)^\top \right]. \tag{8}$$

The computation of this matrix only uses unlabeled data. Since $\Sigma \in \mathbb{R}^{k \times k}$ is a low-dimensional matrix for $k$ not too large, we can accurately estimate it using finite samples from $P_{\mathcal{X}}$. For the ease of theoretical analysis, we assume that we can compute this matrix exactly. Now we define a family of linear heads on the target domain: for $t \in \mathbb{Z}^+$, define

$$g_t(x) = \arg\max_{i \in [r]} \langle f(x), \Sigma^{t-1} b_i \rangle. \tag{9}$$

The case when $t = 1$ corresponds to the linear head in equation (7). When $t$ is large, $g_t$ will care more about the correlation between $f(x)$ and $b_i$ in those directions where the representation variance is large. Intuitively, directions with larger variance tend to contain information also in a more robust way, hence the preconditioner has a "de-noising" effect. See Section 4 for more on why the preconditioning improve the target error. Algorithm 1 gives the pseudocode for this linear classification algorithm.

---

**Algorithm 1** Preconditioned feature averaging (PFA)

---

**Require:** Pre-trained representation extractor $f$, unlabeled data $P_{\mathcal{X}}$, source domain labeled data $\mathcal{P}_S$, target domain test data $\tilde{x}$, integer $t \in \mathbb{Z}^+$
1: Compute the preconditioner matrix $\Sigma := \mathbb{E}_{x \sim P_{\mathcal{X}}} \left[ f(x) f(x)^\top \right]$.
2: **for** every class $i \in [r]$ **do**
3:     Compute the mean feature of the class $i$: $b_i := \mathbb{E}_{(x,y) \sim \mathcal{P}_S} \left[ \mathbb{1}\left[ y = i \right] \cdot f(x) \right]$.
4: **return** prediction $\arg\max_{i \in [r]} \langle f(x), \Sigma^{t-1} b_i \rangle$.

---

We note that this linear head is different from prior work [Shen et al., 2022] where the linear head is trained with logistic loss. We made this modification since this head is more amenable to theoretical analysis. In Section 5 we show that this linear head also achieves superior empirical performance.

The error of a head $g$ on the target domain is defined as:

$$\mathcal{E}_T(g) = \mathop{\mathbb{E}}_{x \sim \mathcal{P}_T} \left[ \mathbb{1}\left[ x \notin T_{g(x)} \right] \right]. \tag{10}$$

The following theorem (proved in Appendix E) shows that the linear head $g_t$ achieves high accuracy on the target domain with a properly chosen $t$:

---

[4]Formally, we have $\mathcal{P}_S(x) := \frac{w(x)}{w(S)} \cdot \mathbb{1}\left[ x \in S \right]$, and $\mathcal{P}_T(x)$ is defined similarly.

[5]We assume access to independent samples from $\mathcal{P}_S$ and thus $b_i$ can be accurately estimated with finite labeled samples in the source domain.

[6]We note that few-shot learning algorithms do not necessarily consider domain shift settings.

**Theorem 3.2.** *Suppose that Assumption 2.2 and 3.1 holds, $P_\mathcal{X}(S)/P_\mathcal{X}(T) \leq O(1)$. Let $f$ be a minimizer of the contrastive loss $\mathcal{L}_2(\cdot)$ and the head $g_t$ be defined in (9). Then, for any $1 \leq t \leq \rho/(8\alpha^2)$, we have*

$$\mathcal{E}_T(g_t) \lesssim \frac{r}{\alpha^2 \lambda_{k+1}^2} \cdot \exp(-\frac{1}{2} t \lambda_{k+1}), \tag{11}$$

*where $\lambda_{k+1}$ is the $k$+1-th smallest eigenvalue of the Laplacian of the positive-pair graph. Furthermore, suppose Assumption 2.3 also holds and $k \geq 2m$, with $t = \rho/(8\alpha^2)$, we have*

$$\mathcal{E}_T(g_t) \lesssim \frac{r}{\alpha^2 \gamma^4} \cdot \exp\left(-\Omega\left(\frac{\rho\gamma^2}{\alpha^2}\right)\right). \tag{12}$$

To see that RHS of equation (12) implies small error, one can consider a reasonable setting where the intra-cluster conductance is on the order of constants (i.e., $\gamma \geq \Omega(1)$). In this case, so long as $\rho \gg \alpha^2 \log(r/\alpha)$, we would have error bound $\mathcal{E}_T(g_t) \ll 1$. In general, as long as $\gamma \gg \alpha^{1/2}$ (the intra-cluster conductance is much larger than cross-cluster connections or its square root) and $\rho$ is comparable to $\alpha$, we have $\rho\gamma^2 \gg \alpha^2$ and thus a small upper bound of the error.

Theorem 3.2 shows that the error decreases as $t$ increases. Intuitively, the PFA algorithm can be thought of as computing a low-rank approximation of a "smoothed" graph with normalized adjacency matrix $\bar{A}^t$, where $\bar{A}$ is the normalized adjacency matrix of the original positive-pair graph. A larger $t$ will make the low-rank approximation of $\bar{A}^t$ more accurate, hence a smaller error. However, there's also an upper bound $t \leq \rho/(8\alpha^2)$, since when $t$ is larger than this limit, the graph would be smoothed too much, hence the corresponding relationship in the graph between source and target classes would be erased. A more formal argument can be found in Section 4.

We also note that our theorem allows "overparameterization" in the sense that a larger representation dimension $k$ always leads to a smaller error bound (since $\lambda_{k+1}$ is non-decreasing in $k$). Moreover, our theorem can be easily generalized to the setting where only polynomial samples of data are used to train the representations and the linear head, assuming the realizability of the function class.

## 3.1 Linear transferability with average relative expansion

In this section, we relax Assumption 3.1 and only assume that the *total connections* from $T_i$ to $S_i$ is larger than that from $T_i$ to $S_j$, formalized below.

**Assumption 3.3** (Average relative expansion (weaker version of Assumption 3.1)). *For some sufficiently large $\tau > 0$, we assume that*

$$\forall i, \ \phi(T_i, S_i) \geq \tau \cdot \alpha^2 \quad \text{and} \quad \forall i \neq j, \ \phi(T_i, S_i) \geq \tau \cdot \phi(T_i, S_j) \tag{13}$$

The following theorem (proved in Appendix F) generalizes Theorem 3.2 in this setting.

**Theorem 3.4.** *Suppose Assumptions 2.2, 2.3 and 3.3 hold, $P_\mathcal{X}(S)/P_\mathcal{X}(T) \leq O(1)$, and feature dimension $k \geq 2m$. Then, for some $t = \Omega\left(\frac{1}{\gamma^2} \cdot \log\left(\frac{1}{\alpha}\right)\right)$, we have*

$$\mathcal{E}_T(g_t) \lesssim \frac{r}{\tau\gamma^8} \cdot \log^2\left(\frac{1}{\alpha}\right). \tag{14}$$

Again, consider a reasonable setting where the intra-cluster conductance is on the order of constants (i.e., $\gamma \geq \Omega(1)$). In this case, so long as $\tau$, the gap between same-class cross-domain connection and cross-class cross-domain connection is sufficiently large (e.g., $\tau \gg r \log^2(1/\alpha)$), we would have an error bound $\mathcal{E}_T(g_t) \ll 1$.

We note that the intra-cluster connections (Assumption 2.3) are necessary, when we only use the average relative expansion (Assumption 3.3 as opposed to Assumption 3.1). Otherwise, there may exist subset $\tilde{T}_i \subset T_i$ that is completely disconnected from $\mathcal{X}\backslash\tilde{T}_i$, hence no linear head trained on the source can be accurate on $\tilde{T}_i$.

# 4 Proof Sketch

**Key challenge:** The analysis will involve careful understanding of how the spectrum of the normalized adjacency matrix of the positive-pair graph is influenced by three types of connections: (i) intra-cluster connections; (ii) connections between same-class cross-domain clusters (between $S_i$ and $T_i$), and (iii) connections between cross-class and cross-domain clusters (between $S_i$ and $T_j$ for $i \neq j$). Type (i) connections have the dominating contribution to the spectrum of the graph, contributing to the top eigenvalues. When analyzing the linear separability of the representations of the clusters, HaoChen et al. [2021] essentially show that type (ii) and (iii) are negligible compared to type (i) connections. However, this paper focuses on the linear transferability, where we need to compare how type (ii) and type (iii) connections influence the spectrum of the normalized adjacency matrix. However, such a comparison is challenging because they are both low-order terms compared to type (i) connections. Essentially, we develop a technique that can take out the influence of the type (i) connections so that they don't negatively influence our comparisons between type (ii) and type (iii) connections.

Below we give a proof sketch of a sligthly weaker version of Theorem 3.2 under a simplified setting. First, we assume $r = 2$, that is, there are two source classes $S_1$ and $S_2$, and two target classes $T_1$ and $T_2$. Second, we assume the marginal distribution over $x$ is uniform, that is, $w(x) = 1/N$ as this case typically capture the gist of the problem in spectral graph theory. Third, we will consider the simpler case where the normalized adjacency matrix $\bar{A}$ is PSD, and the regularization strength $\sigma = 1$.

Let $\tilde{f}(x) = \sqrt{w(x)} \cdot f(x)$ and $\widetilde{F} \in \mathbb{R}^{N \times k}$ be the matrix with $\tilde{f}(x)$ on its $x$-th row. HaoChen et al. [2021] (or Proposition C.1) showed that matrix $\widetilde{F}\widetilde{F}^\top$ contains the top-$k$ eigenvectors of $\bar{A}$. We will first give a proof for the case where $\widetilde{F}\widetilde{F}^\top$ exactly (Section 4.1) or near exactly (Section 4.2) recovers $\bar{A}$. Then we'll give a proof for the more realistic case where $\widetilde{F}\widetilde{F}^\top$ is not guaranteed to approximate $\bar{A}$ accurately (Section 4.3).

## 4.1 Warmup case: when $k = \infty$ and $\widetilde{F}\widetilde{F}^\top = \bar{A}$

In this extremely simplified setting, the inner product between the embeddings perfectly represents the graph (that is, $\langle \tilde{f}(x), \tilde{f}(x') \rangle = \bar{A}_{x,x'}$). As a result, the connections between subsets of vertices, a graph quantity, can be written as a linear algebraic quantity involving $\widetilde{F}$:

$$w(A, B) = \frac{1}{N} \cdot \mathbf{1}_A^\top \bar{A} \mathbf{1}_B = \frac{1}{N} \cdot \mathbf{1}_A^\top \widetilde{F}\widetilde{F}^\top \mathbf{1}_B \tag{15}$$

where $\mathbf{1}_A \in \{0, 1\}^N$ is the indicator vector for the set $A$,[7] and we used the assumption $w(x) = 1/N$.

We start by considering the simple linear classifier which computes the difference between the means of the representations in two clusters.

$$v = \mathbb{E}_{x \sim S_1}[f(x)] - \mathbb{E}_{x \sim S_2}[f(x)] = \widetilde{F}^\top(\mathbf{1}_{S_1} - \mathbf{1}_{S_2}) \in \mathbb{R}^k \tag{16}$$

This classifier corresponds to the head $g_1$ defined in Section 3,[8] which suffices for the special case when $\widetilde{F}\widetilde{F}^\top = \bar{A}$. Applying $v$ to any data point $x \in T_1 \cup T_2$ results in the output $\hat{y}(x) = f(x)^\top v$. For notational simplicity, we consider $\hat{\tilde{y}}(x) = \tilde{f}(x)^\top v = \sqrt{w(x)} f(x)^\top \widetilde{F}^\top (\mathbf{1}_{S_1} - \mathbf{1}_{S_2})$. Because $\hat{y}(x)$ and $\hat{\tilde{y}}(x)$ has the same sign, it suffice to show that $\hat{\tilde{y}}(x) > 0$ for $x \in T_1$ and $\hat{\tilde{y}}(x) < 0$ for $x \in T_2$. Using equation (15) that links the linear algebraic quantity to the graph quantity,

$$\hat{\tilde{y}}(x) = \mathbf{1}_x^\top \widetilde{F}\widetilde{F}^\top(\mathbf{1}_{S_1} - \mathbf{1}_{S_2}) = \mathbf{1}_x^\top \bar{A}(\mathbf{1}_{S_1} - \mathbf{1}_{S_2}) = N \cdot (w(x, S_1) - w(x, S_2)) \tag{17}$$

In other words, the output $\hat{\tilde{y}}$ depends on the relative expansions from $x$ to $S_1$ and from $x$ to $S_2$. By Assumption 3.1 or Assumption 3.3, we have that when $x \in T_1$, $x$ has more expansion to $S_1$ than $S_2$, and vice versa for $x \in T_2$. Formally, by Assumption 3.1, we have that

$$\forall x \in T_1, \ \phi(x, S_1) \geq \rho \gtrsim \phi(x, S_2) \text{ and } \forall x \in T_2, \ \phi(x, S_2) \geq \rho \gtrsim \phi(x, S_1) \tag{18}$$

Because $\phi(x, S_i) = w(x, S_i)/w(x) = N \cdot w(x, S_i)$, we have for $x \in T_1$, $w(x, S_1) > w(x, S_2)$, and therefore by equation (17), $\hat{\tilde{y}}(x) > 0$. Similary when $x \in T_2$, $\hat{\tilde{y}}(x) < 0$.

---

[7]Formally, we have $(\mathbf{1}_A)_x = 1$ iff $x \in A$.

[8]Here because of the binary setting, the classifier can only involve one weight vector $v$ in $\mathbb{R}^d$; this is equivalent to using two linear heads and then compute the maximum as in equation (7).

### 4.2 When $k \ll N$ and $\bar{A}$ is almost rank-$k$

Assuming $k = \infty$ is unrealistic since in most cases the feature is low-dimensional, i.e., $k \ll N$. However, so long as $\bar{A}$ is almost rank-$k$, the above argument still works with minor modification. More concretely, suppose $\bar{A}$'s $(k+1)$-th largest eigenvalue, $1 - \lambda_{k+1}$, is less than $\epsilon$. Then we have $\|\bar{A} - \widetilde{F}\widetilde{F}^\top\|_{\text{op}} = 1 - \lambda_{k+1} \leq \epsilon$. It turns out that when $\epsilon \ll 1$, we can straightforwardly adapt the proofs for the warm-up case with an additional $\epsilon$ error in the final target performance. The error comes from second step of equation (17).

### 4.3 When $\bar{A}$ is far from low-rank

Unfortunately, a realistic graph's $\lambda_{k+1}$ is typically not close to 1 when $k \ll N$ (unless there's very strong symmetry in the graph as those cases in Shen et al. [2022]). We aim to solve the more realistic and interesting case where $\lambda_{k+1}$ is a relatively small constant, e.g., $1/3$ or inverse polynomial in $d$. The previous argument stops working because $\widetilde{F}\widetilde{F}^\top$ is a *very noisy* approximation of $\bar{A}$: the error $\|\bar{A} - \widetilde{F}\widetilde{F}^\top\|_{\text{op}} = 1 - \lambda_{k+1}$ is non-negligible and can be larger than $\|\widetilde{F}\widetilde{F}^\top\|_{\text{op}} = \lambda_k$. Our main approach is considering the power of $\bar{A}$, which reduces the negative impact of smaller eigenvalues. Concretely, though $\|\bar{A} - \widetilde{F}\widetilde{F}^\top\|_{\text{op}} = 1 - \lambda_{k+1}$ is non-negligible, $(\widetilde{F}\widetilde{F}^\top)^t$ is a much better approximation of $\bar{A}^t$:

$$\|\bar{A}^t - (\widetilde{F}\widetilde{F}^\top)^t\|_{\text{op}} = (1 - \lambda_{k+1})^t = \epsilon \tag{19}$$

when $t \geq \Omega(\log(1/\epsilon))$. Inspired by this, we consider the transformed linear classifier

$$v' = \Sigma^{t-1}\widetilde{F}^\top(\mathbf{1}_{S_1} - \mathbf{1}_{S_2}), \tag{20}$$

where $\Sigma = \widetilde{F}^\top\widetilde{F}$ is the covariance matrix of the representations. Intuitively, multiplying $\Sigma$ forces the linear head to pay more attention to those large-variance directions of the representations, which are potentially more robust. The classifier outputs the following on a target datapoint $x$ (with a rescaling of $\sqrt{w(x)}$ for convenience)

$$\hat{y}'(x) = \sqrt{w(x)}f(x)^\top v = \mathbf{1}_x^\top \widetilde{F}\Sigma^{t-1}\widetilde{F}^t(\mathbf{1}_{S_1} - \mathbf{1}_{S_2})$$
$$= \mathbf{1}_x^\top(\widetilde{F}\widetilde{F}^\top)^t(\mathbf{1}_{S_1} - \mathbf{1}_{S_2}) \approx \mathbf{1}_x^\top \bar{A}^t(\mathbf{1}_{S_1} - \mathbf{1}_{S_2}) \tag{21}$$

where the last step uses equation (19). Thus, to understand the sign of $\hat{y}'(x)$, it suffices to compare $\mathbf{1}_x^\top \bar{A}^t \mathbf{1}_{S_1}$ with $\mathbf{1}_x^\top \bar{A}^t \mathbf{1}_{S_2}$. In other words, it suffices to prove that for $x \in T_1$, $\mathbf{1}_x^\top \bar{A}^t \mathbf{1}_{S_1} > \mathbf{1}_x^\top \bar{A}^t \mathbf{1}_{S_2}$.

We control the quantity $\mathbf{1}_x^\top \bar{A}^t \mathbf{1}_{S_1}$ by leveraging the following connection between $\bar{A}$ and a random walk on the graph. First, let $D = \text{diag}(w)$ be the diagonal matrix with $D_{xx} = w(x)$, $A \in \mathbb{R}^{N \times N}$ be the adjacency matrix, i.e., $A_{xx'} = w(x, x')$. Observe that $AD^{-1}$ is a transition matrix that defines a random walk on the graph, and $(AD^{-1})^t$ correspond to the transition matrix for $t$ steps of the random walk, denoted by $x_0, x_t, \ldots, x_t$. Because $\bar{A}^t = (D^{-1/2}AD^{-1/2})^t = D^{1/2}(D^{-1}A)^tD^{-1/2}$ and $D = 1/N \cdot I_{N \times N}$, we can verify that $\mathbf{1}_x^\top \bar{A}^t \mathbf{1}_{S_1} = \Pr[x_t \in S_1 \mid x_0 = x]$. That is, $\mathbf{1}_x^\top \bar{A}^t \mathbf{1}_{S_1}$ and $\mathbf{1}_x^\top \bar{A}^t \mathbf{1}_{S_2}$ are the probabilities to arrive at $S_1$ and $S_2$, respectively. form $x_0 = x$. Therefore, to prove that $\mathbf{1}_x^\top \bar{A}^t \mathbf{1}_{S_1} - \mathbf{1}_x^\top \bar{A}^t \mathbf{1}_{S_2} > 0$ for most $x \in T_1$, it suffices to prove that a $t$-step random walk starting from $T_1$ is more likely to arrive at $S_1$ than $S_2$. Intuitively, because $T_1$ has more connections to $S_1$ than $S_2$, hence a random walk starting from $T_1$ is more likely to arrive at $S_1$ than at $S_2$. In Section E, we prove this by induction.

## 5 Simulations

We empirically show that our proposed Algorithm 1 achieves good performance on the unsupervised domain adaptation problem. We conduct experiments on BREEDS [Santurkar et al., 2020]—a dataset for evaluating unsupervised domain adaptation algorithms (where the source and target domains are constructed from ImageNet images). For pre-training, we run the spectral contrastive learning algorithm [HaoChen et al., 2021] on the joint set of source and target domain data. Unlike the previous convention of discarding the projection head, we use the output after projection MLP as representations, because we find that it significantly improves the performance (for models learned by spectral contrastive loss) and is more consistent with the theoretical formulation. Given the

pre-trained representations, we run Algorithm 1 with different choices of $t$. For comparison, we use the linear probing baseline where we train a linear head with logistic regression on the source domain. The table below lists the test accuracy on the target domain for Living-17 and Entity-30—two datasets constructed by BREEDS. Additional details can be found in Section A.

|           | Linear probe | PFA (ours, $t = 1$) | PFA (ours, $t = 2$) |
|-----------|--------------|---------------------|---------------------|
| Living-17 | 54.7         | 67.4                | 72.0                |
| Entity-30 | 46.4         | 62.3                | 65.1                |

Our experiments show that Algorithm 1 achieves better domain adaptation performance than linear probing given the pre-trained representations. When $t = 1$, our algorithm is simply computing the mean features of each class in the source domain, and then using them as the weight of a linear classifier. Despite having a lower accuracy than linear probing on the source domain (see section A for the source domain accuracy), this simple algorithm achieves much higher accuracy on the target domain. When $t = 2$, our algorithm incorporates the additional preconditioner matrix into the linear classifier, which further improves the domain adaptation performance. We note that our results on Entity-30 is better than Shen et al. [2022] who compare with many state-of-the-art unsupervised domain adaptation methods, suggesting the superior performance of our algorithm.

## 6 Conclusion

In this paper, we study the linear transferability of contrastive representations, propose a simple linear classifier that can be directly computed from the labeled source domain, and prove that this classifier transfers to target domains when the positive-pair graph contains more cross-domain connections between the same class than cross-domain connections between different classes. We hope that our study can facilitate future theoretical analyses of the properties of self-supervised representations and inspire new practical algorithms.

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
