# OpenReview forum: "Beyond Separability: Analyzing the Linear Transferability of Contrastive Representations to Related Subpopulations"
_NeurIPS.cc/2022/Conference — NeurIPS 2022 Accept_

### Official Review · Reviewer_RBh1 · 2022-07-08

**Rating:** 4
**Confidence:** 3
**Soundness:** 2 fair
**Presentation:** 1 poor
**Contribution:** 1 poor

**Summary:**

The paper proposes a simple linear classification algorithm to be applied on top of a feature representation pre-trained using a standard contrastive learning self-supervised pre-training approach. The algorithm is a simple modification of a prototypical network approach by Snell et al. by using a variant of Mahalanobis distance instead of Eucledian one (the covariance matrix power is discussed but used only as 2-1=1 in the experiments). Theoretical guarantees are attempted, however they rely on many assumptions that are: (a) unrealistic to validate/gain intuition in practice; (b) rely on single instance bounds which are very unstable to single instance outlier noise that for sure exists in any real-world learning problem. A single empirical experiment is provided using the BREEDS dataset comparing to a single baseline - linear probing. The experiment is claimed to be one of unsupervised domain adaptation while in fact is of sub-population shift (the goal towards which BREEDS was designed for).

**Questions:**

please see weaknesses

**Limitations:**

not discussed, in fact, there is no "conclusions" section either

**Strengths And Weaknesses:**

Strength:
* very simple (although not novel) linear classification algorithm
* lots of theoretical derivations
* improves one particular application of linear probing on BREEDS (after pre-training on the entire data source+target via self-supervised method which is optimized for ImageNet and hence in prat for BREEDS which is derived from ImageNet)

Weaknesses:
* the algorithm (protonet + Mahalanobis) is not new
* the bounds are using quantities that do not seem to be practically measurable, nor any attempt is made to explain how to measure them in practice - hence it is hard to see how would they provide useful intuitions or guarantees that could be applied in practice
* the bounds rely on assumptions utilizing single sample "max/min-expansion" bounds - but those seem to be highly unstable in presence of outliers - which are always present in every practical learning task ... so it seems unclear how those will be practical to use
* the empirical evaluation on BREEDS is clearly not UDA (as claimed) it is of sub-population shift (labeled samples are one type of cat and unlableld samples are another type of cat). I also assume the main reason for gains over linear probing are: (a) overfitting of the linear probing, probably better hyper-params could improve it considerably for BREEDS; (b) too small data for self-supervised pre-training of the representation model - probably better to take a well-optimized full ImageNet pre-trained model like DINO or MocoV2
* a single experiment, not well defined and with a single very simplistic baseline (linear probe) does not seem to be sufficient
* writing is not very clear, many experimental details are omitted, and code is neither promised nor provided (claimed proprietary in the checklist) - as it stands the paper does not bring much value in the empirical sense as all that makes its (limited) empirical results not reproducible as it seems

POST Rebuttal:
I must say that even after an in-depth discussion with the authors following their rebuttal response, I am not convinced that this paper should be accepted in its current form. Mainly due to some concerns with the practicality of the proposed theory that in my view would require many additional experiments that are currently not performed (as I explained in detail in my discussion with the authors). I, however, would like to acknowledge the value of author's efforts during the rebuttal, as well as the more positive opinion of the other reviewers, and hence, am raising my score by 1 point to "borderline reject".

---

> ### Author Response · Authors · 2022-08-02
> **Response to Reviewer RBh1 (3/3)**
>
> > "I also assume the main reason for gains over linear probing are: (a) overfitting of the linear probing, probably better hyper-params could improve it considerably for BREEDS; (b) too small data for self-supervised pre-training of the representation model - probably better to take a well-optimized full ImageNet pre-trained model like DINO or MocoV2"
>
> We thank the reviewer for sharing the intuitions. We would like to first reiterate that our main focus is not improving upon linear probing. Instead, our goal is to explain why linear probing on contrastive representation can work, and we study PFA because it’s more amenable to theoretical analysis.
>
> We note that we’ve done hyperparameter search during linear probing, and the linear probing algorithm we used is consistent with prior empirical works [11]. Also, we used the same amount of pre-training data for linear probing and PFA, thus the improvement of PFA shouldn’t come from insufficient pre-training data.
>
> We believe that PFA is better than linear probing because it adjusts the feature according to the population data and leads to better transferability, which is also revealed in our theoretical analysis. In contrast, the trained linear head would find the max-margin solution on the source domain, overfitting to the decision boundary on the source. One empirical evidence is that PFA actually **hurts the test accuracy on the source domain** despite improving the test performance on the target domain compared to linear probing (LP), as shown in the table below. If it’s only due to sample complexity of the source labeled data, PFA should have improved the test accuracy on both the source and target domains simultaneously.
>
> |         | LP (Source)    | PFA (Source)   | LP (Target)   | PFA (Target)   |
> | -------- | ------- |-------- |--------- |-------- |
> | Living-17    | 91.3   | 90.5   | 54.7   | 72.0  |
> | Entity-30   | 84.8   | 77.3    |46.4   |65.1   |
>
> > "a single experiment, not well defined and with a single very simplistic baseline (linear probe) does not seem to be sufficient"
>
> We first want to reiterate that our original goal for including experiments was to just provide complementary results for our theory. Our goal was not to find an algorithm that works better than state-of-the-art methods, so we didn’t test it on many different datasets.
>
> That being said, we thank the reviewers for suggesting adding more datasets and comparing with more baseline methods. We add **comparison with more domain adaptation methods** (ERM, SENTRY [8], DANN [9]). In addition to the two existing datasets living-17 and entity-30, we also add **new results on the STL->CIFAR10 dataset** which is a standard UDA dataset [10]. We also compare with linear probing (LP) on the pre-mlp features of a contrastive learned model.. The results are listed in the following table (details in Appendix A.1):
>
> |       | ERM   | SENTRY   | DANN   | LP (pre-mlp)  |LP (post-mlp)  |PFA (post-mlp)   |
> | ------ | ------ | ------ | ------ | ------ | ------ | ------ |
> | Living-17  | 63.3   | 75.5  | 71.3   | **79.1**  | 54.7   | 72.0  |
> | Entity-30  | 52.5  | 56.1  | 57.5  | 63.8  | 46.4   | **65.1**  |
> | STL->CIFAR10   | 57.4  | 53.8   | 55.2  | 79.8  | 73.1  | **80.0**  |
>
> Our experiments show that PFA consistently improves upon direct linear probing on the post-mlp contrastive representation. Furthermore, PFA is competitive and usually better than other domain adaptation algorithms such as SENTRY and DANN. These results show the practical relevance of our theoretical analysis, but we are not advocating PFA as a new empirical method.
>
> > "code is neither promised nor provided (claimed proprietary in the checklist) "
>
> Thanks for pointing this out, we’ve included code in the updated version of supplementary materials, and will make it public after the camera-ready version.
>
> References:
>
> [1] Connect, not collapse: Explaining contrastive learning for unsupervised domain adaptation. Shen et al. ICML 2022.
>
> [2] Self-training avoids using spurious features under domain shift. Chen et al. NeurIPS 2020.
>
> [3] Understanding self-training for gradual domain adaptation. Kumar et al. ICML 2020.
>
> [4] A theory of learning from different domains. Ben-David et al. Machine learning 2010.
>
> [5] Covariate shift adaptation by importance weighted cross validation. Sugiyama et al. JMLR
> 2007.
>
> [6] Bridging theory and algorithm for domain adaptation. Zhang et al. ICML 2019.
>
> [7] On learning invariant representations for domain adaptation. Zhao et al. ICML 2019.
>
> [8] Selective entropy optimization via committee consistency for unsupervised domain adaptation. Prabhu et al. ICCV 2021.
>
> [9] Domain-adversarial training of neural networks. Ganin et al. JMLR 2016.
>
> [10] A dirt-t approach to unsupervised domain adaptation. Shu et al. ICLR 2018.
>
> [11] A simple framework for contrastive learning of visual representations. Chen et al. ICML 2020.

---

> > ### Comment · Reviewer_RBh1 · 2022-08-05
> > **Re - additional experiments**
> >
> > thank you for these additional experiments, looking at the original BREEDS paper page 32 (https://openreview.net/pdf?id=mQPBmvyAuk) I see much higher numbers for linear probing on Living-17 / Entity-30 (it is called Target-RT there). I believe those are better put in the additional table you reported above and in this case, would not favor PFA as these numbers are at least 7% - 10% higher for both datasets depending on a parameter there. I also do not see any response to my concern about applying SSL on small data, trying PFA vs LP over established SOTA SSL models would be more useful to the community in my opinion. Also, I am still not convinced that the author's experiment on BREEDS fits the UDA paradigm and that UDA methods (btw, the compared methods are not UDA SOTA) are the best baselines here. At best, the differences between different sub-classes in BREEDS are subtle (fine-grained) and all the images come from the real domain so features are easily transferable.
> > I am still not sufficiently convinced, sorry.

---

> > > ### Author Response · Authors · 2022-08-06
> > > **Thank you for your response**
> > >
> > > We thank the reviewer for their response!
> > >
> > > Regarding the “Target-RT” method in the original BREEDS paper, we note that their method requires retraining the last linear layer with **labeled** target domain data, whereas we don’t use any labeled target domain data. Thus, it’s unfair to compare the “Target-RT” number with our number. In fact, our number is higher than the “Target” column in the BREEDS paper where no labeled target data is used.
> > >
> > > We agree that adding more unlabeled data in pre-training can improve the performance. However, we believe that’s orthogonal to the problem we study, which is to achieve good performance on the target domain given only labeled source data and unlabeled target data.
> > >
> > > Regarding UDA, in this paper we use UDA to refer to settings where the source domain and target domain have different distributions but the same label set, thus both living-17 and entity-30 fall under this term. We’ve also added another domain adaptation dataset (STL->CIFAR10), which is a benchmark dataset that has been used in many previous works that study UDA (e.g., [1]).
> > >
> > > [1] A dirt-t approach to unsupervised domain adaptation. Shu et al. ICLR 2018.

---

> > > > ### Comment · Reviewer_RBh1 · 2022-08-08
> > > > **not so accurate...**
> > > >
> > > > Well, in BREEDS the target data is not used for training in any form, while afaik, you did use it for your pre-training.
> > > > So if you would really want to stick to the DA terminology, then what you do would be like UDA and what BREEDS report in their Target performance would be UDG (= Unsupervised Domain Generalization = testing on completely unseen target domain). I think properly comparing to BREEDS would require you to implement UDG for your method.

---

> > > > > ### Author Response · Authors · 2022-08-09
> > > > > **Response to "not so accurate..."**
> > > > >
> > > > > We apologize for the confusion, but just wanted to take this chance to clarify how we used the BREEDS datasets and how prior works used the dataset. The BREEDS datasets (Living-17 and Entity-30) contain two domains, a source domain and a target domain.
> > > > >
> > > > > - Our paper: PFA **and the baselines** DANN and SENTRY, all make use of labeled source examples and unlabeled target examples. So we compare PFA and the baselines in a fair way.
> > > > > - Prior work [1] has also used BREEDS in this way: they benchmark methods that use labeled source examples, and unlabeled target examples.
> > > > > - [2], the original BREEDS paper, consider two methods. Method 1 is standard ERM, which only uses labeled source examples. Method 2 is target-RT, which uses labeled target examples. See page 7 of the original BREEDS paper(https://openreview.net/pdf?id=mQPBmvyAuk): "we retrain the last (fully-connected) layer of models trained on the source domain with data from the target domain".
> > > > >
> > > > > The reviewer is correct that what we do is like UDA, which is why we compare to other UDA baselines (DANN and SENTRY) which use source labeled data and target unlabeled data. We will clarify these settings - thank you to the reviewer for the suggestion.
> > > > >
> > > > > In addition, we’ve also included additional experiments on STL->CIFAR10, which is a more common UDA dataset in previous papers that study UDA (e.g., [2, 3]). Our experiment results on STL->CIFAR10 are consistent with that on Living-17 and Entity-30.
> > > > >
> > > > > References:
> > > > >
> > > > > [1] Connect, Not Collapse: Explaining Contrastive Learning for Unsupervised Domain Adaptation. Shen et al. ICML 2022.
> > > > >
> > > > > [2] A dirt-t approach to unsupervised domain adaptation. Shu et al. ICLR 2018.
> > > > >
> > > > > [3] Self-ensembling for visual domain adaptation. French et al. ICLR 2018.

---

> > > > > > ### Comment · Reviewer_RBh1 · 2022-08-09
> > > > > > **to make it stronger**
> > > > > >
> > > > > > I would suggest testing on datasets like DomainNet and VisDA - those are much larger scale and common for SOTA UDA testing

---

> > > > > > > ### Author Response · Authors · 2022-08-09
> > > > > > > **Thanks for the suggestion**
> > > > > > >
> > > > > > > We thank the reviewer for suggesting testing on larger scale datasets. At the moment, we are running experiments on them and will add those results to the camera ready version.

---

> ### Author Response · Authors · 2022-08-02
> **Response to Reviewer RBh1 (2/3)**
>
> > "the bounds are using quantities that do not seem to be practically measurable, nor any attempt is made to explain how to measure them in practice" "the bounds rely on assumptions utilizing single sample 'max/min-expansion' bounds - but those seem to be highly unstable in presence of outliers - which are always present in every practical learning task"
>
> Our main result (Theorem 3.4, proofs in Appendix F) only **requires average expansion** (Assumption 3.3), and **does not require single sample "max/min-expansion" bounds**. We also state a special case (Theorem 3.2) as a warm-up which assumes min/max expansion, for simplicity of exposition. Indeed, average expansion is more stable in the presence of outliers. We’d like to apologize for this miscommunication, and we’ll make this point more clear in the next version of the paper.
>
> [1] already contains experiments that test the average inter-cluster expansion (which they call “connectivity”). Concretely, for any given pair of clusters (e.g., two classes from two different domains), they train a binary classifier to classify which cluster an augmented image belongs to. The test accuracy on fresh augmented images is regarded as a measure of the strength of expansion/connectivity between the two clusters. Their results show that the same class in two domains has larger expansion than two different classes in two domains, which corresponds to our Assumption 3.3.
>
> The reviewer seems to have doubts about whether our theoretical assumptions are too strong. In fact, **our assumptions are weaker than previous theoretical papers** that study domain adaptation. For instance, one line of prior works [2,3] assume that data is Gaussian or near-Gaussian. Another line of work requires bounded density ratios or overlap between the source and target domains [4,5,6,7], but our theory allows for non-overlapping source and target domains. We also note that in practice one can always increase expansion by applying stronger data augmentation, so we believe that assuming an expansion lower bound is consistent with practical settings.
>
> > "The empirical evaluation on BREEDS is clearly not UDA (as claimed) it is of sub-population shift (labeled samples are one type of cat and unlabeled samples are another type of cat). "
>
> We believe that UDA means one has labeled data from a source domain and unlabeled data from a target domain, and the goal is to get high accuracy on the target domain. Although the original BREEDS dataset is claimed to be about subpopulation shift, the two datasets we use — living-17 and entity-30 — are in fact UDA tasks created from BREEDS, where source and target domains contain non-overlapping ImageNet classes. For instance, for the label “ape” in living-17, the source and target domains may contain ImageNet data from “gibbon” and “gorilla”, respectively. Thus, the reviewer is correct about the nature of the dataset — indeed the source and target domains share the same class labels (e.g., ape, or cat, as the reviewer mentioned), but the support of the two domains are largely disjoint.
>
> That being said, we are more than willing to change the naming if the reviewer believes there are other choices of words that would make our statements more accurate.

---

> > ### Comment · Reviewer_RBh1 · 2022-08-05
> > **regarding not requiring max/min-expansion bounds for "main result"**
> >
> > Theorem 3.4 (page 7, line 259) states at its very beginning: "Suppose Assumptions 2.2, 2.3 and3.3 hold...". Now looking at Assumption 2.2 (page 4, line 142) we see the max expansion. So it seems to require that, right?
> > Regarding BREEDS and UDA please see my comment on the next point below.
> > Sorry, still not convinced...

---

> > > ### Author Response · Authors · 2022-08-06
> > > **Thank you for your reponse**
> > >
> > > We thank the reviewer for their response!
> > >
> > > We apologize for not being clear in our response. By not requiring “max/min-expansion bounds” we meant that we don’t require Assumption 3.1 which states that every data in the source domain has more connections to the corresponding class in the target domain class than other classes in the target domain. This is the most restrictive assumption for our Theorem 3.2, but we were able to relax it to the average case in Theorem 3.4. Indeed, Assumption 2.2 still assumes an upper bound on data-wise max-expansion. However, we believe this is a reasonable assumption (at least for image data): say we have two classes ‘dog’ and ‘cat’ in the population, and one data represents a dog. Under natural augmentations (cropping, flipping, blurring, etc), it’s reasonable that the vast majority of its augmentations are still a dog.
> > >
> > > We note that there is no prior work theoretically studying why self-supervised learning helps domain adaptation. In this work, we provide the **first theoretical analysis** that shows contrastive learning helps unsupervised domain adaptation in a general setting. We believe that making mild assumptions on data and building theory on top of them is a reasonable route toward understanding this empirical mystery.

---

> > > > ### Comment · Reviewer_RBh1 · 2022-08-08
> > > > **still consistent with my original concern**
> > > >
> > > > so given that understanding, Assumption 2.2 still requires there should be NO outliers (as you require \alpha << 1 in line 143).
> > > > Every class has samples that are on the boundary with other classes, e.g. in your cat vs dog example, some dog can look like a cat up to a small detail (e.g. dog face) and then an augmentation (e.g. random masking) could remove that difference producing an outlier. This is ofc one example, but importantly - outliers always exist and it was not clear to me how much of your theory is robust to that.

---

> > > > > ### Author Response · Authors · 2022-08-09
> > > > > **Response to “still consistent with my original concern”**
> > > > >
> > > > > We thank the reviewer for raising this insightful question. In fact, **our theory is robust to ''outliers'' in the data**. We’ve included an additional theoretical result (Theorem G.2) in our latest revision, which only requires the data to have a clustering structure after removing certain ''outliers'' (Assumption G.1). Roughly speaking, so long as the amount of positive pairs that contain ''outliers'' is less than the amount of positive pairs that connect the corresponding source-target classes, the PFA algorithm can provably achieve good accuracy on the target domain. Please see Appendix G for more details.
> > > > >
> > > > > Optional details: We note that, to some extent, this is arguably the minimal requirement for any domain adaptation method to have good performance on the target (without making additional structure assumptions on ''outliers''). The reason is as follows: suppose the amount of connection to outliers is larger than the amount of connection between source-target classes, one can construct adversarial outliers that connect one target class $T_i$ to an incorrect source class $T_j$, and any reasonable domain adaptation method would fail on $S_i$.
> > > > >
> > > > > We thank the reviewer for the suggestion, and believe this makes our paper stronger. We will add this theorem and its discussion to the main body of the paper when given additional space for camera ready.

---

> ### Author Response · Authors · 2022-08-02
> **Response to Reviewer RBh1 (1/3)**
>
> We thank Reviewer RBh1 for the detailed and useful suggestions.
>
> We believe that the reviewer’s evaluation of the paper focused on our experiments. To clarify, the focus of our paper is the **theoretical understanding** of the phenomenon, instead of proposing any new practical methods. Prior empirical works such as [1] have observed that self-supervised features achieve state-of-the-art performance on many unsupervised domain adaptation benchmark datasets, but it remained unclear why they work. Therefore, our goal is to **theoretically** analyze how linear probing on self-supervised features solve unsupervised domain adaptation. As far as we know, we provide the **first** theoretical analysis that shows contrastive learning helps domain adaptation in a general setting (beyond the toy examples that are studied in [1]).
>
> The experiment section is mostly for the complementary purpose of demonstrating the practical relevance of our theory. Our theoretical analysis provides guarantees for the PFA algorithm, a variant of linear probe. We study PFA because it enjoys nice theoretical properties, resembles existing methods, and is more amenable to theoretical analysis. Thus, our focus is the theoretical analysis of PFA, and we reported the empirical performance only as a supplementary component to show the practical relevance of the PFA algorithm. Somewhat surprisingly, PFA indeed achieves competitive performance for some UDA benchmark datasets even with post-mlp features, which may be of independent interest to people in a broader empirical community.
>
> We very much appreciate reviewers’ suggestions on adding more datasets and comparing with more baseline methods. We have included additional experiments in the revision and response below. However, we also respectfully ask the reviewer to mostly consider the theoretical contribution of the paper in the evaluation.
>
> > The reviewer believes that “the algorithm (protonet + Mahalanobis) is not new”
>
> As discussed above, our **main contribution is the theoretical analysis** of how contrastive learning helps domain adaptation, instead of proposing any new practical methods. We study the PFA algorithm because it is more amenable for theoretical analysis and resembles existing methods such as linear probing. As far as we know, we are the first to **prove** that this simple algorithm works well as a domain adaptation algorithm, which a priori might  be surprising to the community.

---

> > ### Comment · Reviewer_RBh1 · 2022-08-05
> > **I agree theory is valuable but...**
> >
> > I feel that it (the theory) is most valuable when you can use it to predict some practical phenomena. So, by no means I do not want to discourage the authors from continuing their exciting work. I just feel that the paper in its current state would be much less impactful to the community as opposed if the authors would use it to discover some unknown fact (the fact that LP or variants work well on top of self-supervised learning is already known fact as the authors admit) or one could use it to derive some novel insights that could, in turn, be converted to some practical novel method. Unfortunately, I still do not see any of this in the current version of the paper. However, I could be wrong...

---

> > > ### Author Response · Authors · 2022-08-06
> > > **Thank you for your response**
> > >
> > > We thank the reviewer for sharing their thoughts on the value of theory!
> > >
> > > We note that our theory indeed predicts several phenomena that are verified by experiments. For instance, our theory suggests that adding a simple preconditioner (which is the covariance matrix of the SSL representation) can improve the domain adaptation performance, which is also observed in our experiments. Our theory also suggests that using post-mlp features from SSL methods could work well for domain adaptation, which is indeed empirically true.
> > >
> > > We agree that designing better practical methods is important, though we believe that it goes beyond the scope of this work. The main contribution of our theory is developing theoretical tools (e.g., the augmentation graph perspective) to explain how existing algorithms work, which we believe by itself is an important and challenging question.

---

> > > > ### Comment · Reviewer_RBh1 · 2022-08-08
> > > > **to increase the impact**
> > > >
> > > > I would suggest that if you are confident in what your theory predicts, then to make this paper much more impactful - it would be great to see more UDA experiments on many standard UDA benchmarks you are currently not testing on. As I mentioned below, BREEDS was not intended for that, and the original BREEDS paper did not provide any baseline on BREEDS for the UDA setting, neither did any other work that you tested, nor UDA SOTA which you have not.

---

> > > > > ### Author Response · Authors · 2022-08-09
> > > > > **Response to “to increase the impact”**
> > > > >
> > > > > As we mentioned in our response below, we’ve added experiments on STL->CIFAR10, which is a standard UDA dataset used by prior work. That being said, we want to reiterate that experiments are only complementary to our core contribution, which is developing techniques to theoretically analyze how contrastive learning helps domain adaptation.

---

> ### Author Response · Authors · 2022-08-05
> **Any other questions or suggestions?**
>
> Thank you again for the comprehensive and useful review, and sorry to bother you. We just wanted to check if you had any other questions or suggestions - we would love to get feedback to further improve our work.
>
> So far the review has focused on the experiments part of the paper, and has raised some questions about the limitation of experiments. We hope our response has clarified the focus and novelty of the paper, and has addressed the reviewer's concerns?

---

> ### Comment · Reviewer_RBh1 · 2022-08-08
> **My final score**
>
> I must say that even after an in-depth discussion with the authors following their rebuttal response, I am not convinced that this paper should be accepted in its current form. Mainly due to some concerns with the practicality of the proposed theory that in my view would require many additional experiments that are currently not performed (as I explained in detail in my discussion with the authors). I, however, would like to acknowledge the value of author's efforts during the rebuttal, as well as the more positive opinion of the other reviewers, and hence, am raising my score by 1 point to "borderline reject".

---

> > ### Author Response · Authors · 2022-08-09
> > **Thank you to the reviewer**
> >
> > Thank you to the reviewer for engaging closely with us, and giving us useful suggestions (e.g., adding more experiments and showing that the theory is robust to outliers — which we've done in the latest revision).

---

### Official Review · Reviewer_TUn2 · 2022-07-11

**Rating:** 5
**Confidence:** 4
**Soundness:** 2 fair
**Presentation:** 2 fair
**Contribution:** 2 fair

**Summary:**

The paper aims to provide an analysis of the linear transferability of contrastive learning in a general domain adaptation setting. In particular, they theoretically show that contrastive learning can encode cluster identities and catch the inter-cluster relationship. Experimental results on only one data set show that the proposed preconditioned feature averaging (PFA) provides better results in a comparison with linear probe.

**Questions:**

Please see my comments on the weaknesses of the paper

**Limitations:**

Please see my comments on the weaknesses of the paper

**Strengths And Weaknesses:**

Originality: The authors claim that their paper can provide analyzes of the linear transferability with weaker and more realistic assumptions compared to existing theoretical works. However, to be honest, since I did not have enough time to verify this. I'll be then based on the evaluations from other reviewers.

Quality: My major concern is about the empirical demonstration of the paper. In particular, the experimental results of the paper are indeed poor since they are only evaluated on only one data set (BREEDS) and the proposed method is compared with only one other method (linear problem). Having said that, I think that the authors definitely need more work on the empirical part of the paper since the current version looks like incomplete work.

Clarity: The paper is well-written and easy to follow

Significance: Besides the concern above about the empirical results of the paper, I personally think that only studying the linear transferability in domain adaptation is quite insignificant since it can restrict the application of contrastive learning and domain adaptation in modern deep learning.

---

> ### Author Response · Authors · 2022-08-02
> **Response to Reviewer TUn2**
>
> We thank Reviewer TUn2 for the detailed and useful suggestions, and for saying that our paper is “well-written and easy to follow”.
>
> We believe the reviewer’s evaluation of the paper focused on our experiments. To clarify, the focus of our paper is the **theoretical understanding** of the phenomenon, instead of advocating  any new practical methods. Prior empirical works such as Shen et al. have observed that self-supervised features achieve state-of-the-art performance on  many unsupervised domain adaptation benchmark datasets, but it remained unclear why they work. Therefore, our goal is to **theoretically** analyze how linear probing on self-supervised features solve unsupervised domain adaptation. As far as we know, we provide the **first** theoretical analysis that shows contrastive learning helps unsupervised domain adaptation in a general setting (beyond the toy examples that are studied in [1]).
>
> The experiment section is mostly for the complementary purpose of demonstrating the practical relevance of the theoretical analysis. Our theoretical analysis provides guarantees for the PFA algorithm, a variant of linear probe. We study PFA because it enjoys nice theoretical properties, resembles existing methods, and is more amenable to theoretical analysis. Thus, our focus is the theoretical analysis of PFA, and we reported the empirical performance only as a supplementary component to show the practical relevance of the PFA algorithm. Somewhat surprisingly, PFA indeed achieves competitive performance for some UDA benchmark datasets even with post-mlp features, which may be of independent interest to people in a broader empirical community.
>
> We very much appreciate reviewers’ suggestions on adding more datasets and comparing with more baseline methods. We have included additional experiments in the revision and response below. However, we also respectfully ask the reviewer to mostly consider the theoretical contribution of the paper in the evaluation.
>
> We add comparison with **more domain adaptation methods** (ERM, SENTRY [2], DANN [3]). We also compare with linear probing on the pre-mlp features of a contrastive learned model. In addition to the two existing datasets living-17 and entity-30, we also add **new results on the STL->CIFAR10 dataset** which is a standard UDA dataset [4]. The results are listed in the following table (more details in Appendix A.1):
>
> |       | ERM   | SENTRY   | DANN   | LP (pre-mlp)  |LP (post-mlp)  |PFA (post-mlp)   |
> | ------ | ------ | ------ | ------ | ------ | ------ | ------ |
> | Living-17  | 63.3   | 75.5  | 71.3   | **79.1**  | 54.7   | 72.0  |
> | Entity-30  | 52.5  | 56.1  | 57.5  | 63.8  | 46.4   | **65.1**  |
> | STL->CIFAR10   | 57.4  | 53.8   | 55.2  | 79.8  | 73.1  | **80.0**  |
>
> Our experiments show that PFA consistently improves upon direct linear probing on the post-mlp contrastive representation. Furthermore, spectral contrastive learning + PFA is competitive and usually better than other domain adaptation algorithms such as SENTRY and DANN. These results show the practical relevance of our PFA algorithm (and spectral contrastive learning) and its theoretical analysis.
>
>
> > "Only studying the linear transferability in domain adaptation is quite insignificant since it can restrict the application of contrastive learning and domain adaptation in modern deep learning."
>
> Our understanding is that the reviewer is expecting more analysis on fine-tuning (instead of linear probing). We agreed that analyzing fine-tuning is an important open question, but it's challenging because the community appears to lack theoretical tools to analyze the optimization trajectory of neural networks from a pretrained model. In fact, there is no prior theoretical work on fine-tuning beyond toy examples (e.g., 2-layer **linear** networks).
>
> We also note that empirically models with better performance under linear probing usually also achieve better fine-tuning performance. Linear probing also has its advantage of being fast and more reliable on out-of-distribution data [5]. Thus, studying linear probing methods can be viewed as a surrogate for understanding fine-tuning.
>
> References:
>
> [1] Connect, not collapse: Explaining contrastive learning for unsupervised domain adaptation. Shen et al. ICML 2022.
>
> [2] Selective entropy optimization via committee consistency for unsupervised domain adaptation. Prabhu et al. ICCV 2021.
>
> [3] Domain-adversarial training of neural networks. Ganin et al. JMLR 2016.
>
> [4] A dirt-t approach to unsupervised domain adaptation. Shu et al. ICLR 2018.
>
> [5] Fine-tuning can distort pretrained features and underperform out-of-distribution. Kumar et al. ICLR 2022.

---

> > ### Comment · Reviewer_TUn2 · 2022-08-06
> > **increase the score to 5**
> >
> > Thank you for answering my questions and providing additional experimental results on domain adaptation. Though I am aware and agree with the authors and other reviewers that this paper might have interesting theoretical analyses, I personally think that the authors should work more on the empirical part to support the theoretical part of the paper, which is claimed to be its main contribution.
> >
> > I decide to upgrade the score of the paper to 5.

---

> > > ### Author Response · Authors · 2022-08-09
> > > **Thank you to the reviewer**
> > >
> > > Thank you to the reviewer for engaging closely with us, and giving us useful suggestions (e.g., adding more experiments — which we've done in the latest revision).

---

> ### Author Response · Authors · 2022-08-05
> **Any other questions or suggestions?**
>
> Thank you again for the comprehensive and useful review, and sorry to bother you. We just wanted to check if you had any other questions or suggestions - we would love to get feedback to further improve our work.
>
> So far the review has raised some concerns on the experiments part of the paper. We hope our response has clarified the focus and novelty of the paper, and has addressed the reviewer's concerns?

---

### Official Review · Reviewer_GwWK · 2022-07-12

**Rating:** 7
**Confidence:** 4
**Soundness:** 4 excellent
**Presentation:** 4 excellent
**Contribution:** 4 excellent

**Summary:**

This paper theoretically studies the previously reported success of contrastive learning on unsupervised domain adaptation, where unlabeled data is available for the source and target domains but labels are only available for the source domain. This involves 3 steps, (1) train representations using contrastive learning on the combined unlabeled data for source and target, (2) train a linear classifier on source representations using source labels, (3) use the linear classifier *directly* for the target representations. This idea works well in some domain adaptation setting and is referred to as *linear transferability* in this paper.

The paper identifies natural conditions under which this procedure can provably works: data from the same class in different domains are more related to each other than data from different classes in different domains. This idea is formalized using the (weighted) augmentation graph introduced in HaoChen et al. where there is an edge between two data points if they appear as a positive pair. The analysis requires 3 main properties to be satisfied by the graph to ensure linear transferability: (a) few edges going outside of any class, (b) high expansion within a class, (c) high expansion across the same class in different domains. Proving this result requires extending the analysis for spectral contrastive learning from HaoChen et al.

The paper also proposes a new algorithm (PFA) to learn the source linear classifier: use the mean representation for classes, multiplied by an appropriate pre-conditioner for a de-noising effect. Experiments on BREEDS (constructed from ImageNet) show that PFA outperforms the baseline of logistic regression.


**Questions:**

Other comments and questions
- Any intuition for why learned classifier (logistic regression) is worse than PFA on target domain? Is it an issue of labeled sample complexity in the source domain or is it more fundamental and would happen even with infinite labeled samples? Perhaps some simple experiments (by varying amount of source labeled samples) could shed some light on this.
- Is there a way to test (directly or indirectly) whether such expansions assumption holds, especially for min and max notions of expansions? It could also be useful to discuss what could happen if one or more of the assumptions do not hold.
- L214: “vulnerable to complex asymmetric structures” could use some clarification
- Near L140 it would be useful to mention that $C_1, \dots, C_m$ will correspond to classes of interest
- L237: Typo in “equation (50)”?
- L263: Should it be $\tau \gg r \log^2(r/\alpha)$?
- Some typos in L616 ($w({\color{red}x’})$ at the end) and equations 40 and 41 ($w({\color{red}x}, \tilde{x})$).


**Limitations:**

Some limitations are discussed. No potential negative societal impact discussed.

**Strengths And Weaknesses:**

Strengths

- The phenomenon of linear transferability is quite interesting, as is the fact that contrastive learning helps with it.
- The paper adequately positions the required assumptions and analysis compared to prior work on domain adaptation and contrastive learning. For e.g., the comparison to Cai et al. using multiple notions of source and target transferability was useful in understanding the effect of different assumptions.
- It is well written and easy to follow. The proof sketches with increasing level of difficulty is a very useful addition to provide intuitions for the analysis.
- I skimmed over some of the proofs in the appendix; they are reasonably well-structured, clearly written and I did not find any obvious mistakes.



Weaknesses

- There is currently no discussion about interesting future directions
- While PFA (mean classifier + preconditioner) works better than logistic regression, there is no justification or intuition provided for this finding. More on this in questions for authors
- While the assumption seems intuitive, no verification for it, experimental or otherwise, is provided


Overall I think this paper is a solid contribution towards better understanding contrastive learning for unsupervised domain adaptation and I vote for accepting this paper.

---

> ### Author Response · Authors · 2022-08-02
> **Response to Reviewer GwWK (2/2)**
>
> > "While the assumption seems intuitive, no verification for it, experimental or otherwise, is provided. Is there a way to test (directly or indirectly) whether such expansions assumption holds, especially for min and max notions of expansions? "
>
> We note that our more general Theorem 3.4 only requires *average* expansion (Assumption 3.3), and does not require single sample "max/min-expansion" bounds. The single sample max/min-expansion (Assumption 3.1) is only assumed for our first Theorem (Theorem 3.2), which is a more restricted case that we single out because its proof is cleaner (as we sketched in Section 4) and captures the essence of the more general setting.
>
> [1] already contains experiments that test the average inter-cluster expansion (which they call “connectivity”). Concretely, for any given pair of clusters (e.g., two classes from two different domains), they train a binary classifier to classify which cluster an augmented image belongs to. The test accuracy on fresh augmented images is regarded as a measure of the strength of expansion/connectivity between the two clusters. Their results show that the same class in two domains has larger expansion than two different classes in two domains, which corresponds to our Assumption 3.3.
>
> > "L214: 'vulnerable to complex asymmetric structures' could use some clarification"
>
> When the positive-pair graph is like a stochastic block model which is very symmetric and has a low rank adjacency matrix, directly using the mean feature without any preconditioner can already provably achieve good test accuracy on the target domain. A proof sketch for this setting can be found at the beginning of Section 4. However, when the adjacency matrix is not low rank (hence has “complex asymmetric structures”), the preconditioner is necessary for the proof. We apologize for the lack of clarification, and will make it more clear in the camera ready version of the paper when given additional space.
>
> > "L237: Typo in 'equation (50)'?" "Near L140 it would be useful to mention that $C_1, C_2, \cdots, C_m$ will correspond to classes of interest" "L263: Should it be $\tau\ge r\log^2(r/\alpha)$?" "Some typos in L616 ($w(x')$ at the end) and equations 40 and 41 ($w(x, \tilde{x})$)"
>
> We thank the reviewer for pointing out the typos and the suggestions on writing. We’ve fixed them in our revision of the paper.
>
> References:
>
> [1] Connect, not collapse: Explaining contrastive learning for unsupervised domain adaptation. Shen et al. ICML 2022.

---

> ### Author Response · Authors · 2022-08-02
> **Response to Reviewer GwWK (1/2)**
>
> We thank Reviewer GwWK for the positive review and useful suggestions, for appreciating that “linear transferability is quite interesting” and for “voting for accepting this paper”.
>
> We address the questions of the reviewers below.
>
> > "There is currently no discussion about interesting future directions"
>
> We thank the reviewer for the suggestion. We hope that our study can facilitate future theoretical analyses of self-supervised representations and inspire new practical algorithms. One interesting direction would be studying how distributional-robust features can be learned in multimodal contrastive learning methods such as CLIP. Another interesting question is whether we can generalize the analysis to incorporate fine-tuning, which usually achieves better downstream performance than linear probing. We will add this to the camera ready version of our paper when given additional space.
>
> > “While PFA (mean classifier + preconditioner) works better than logistic regression, there is no justification or intuition provided for this finding.” “Any intuition for why learned classifier (logistic regression) is worse than PFA on target domain?” “Perhaps some simple experiments (by varying amount of source labeled samples) could shed some light on this”
>
> Thanks for suggesting the experiment! We added a **new experiment** where we **vary the amount of labeled source data**, and found that **PFA consistently improves over learned linear head** (LP) on living-17 and entity-30. The results are listed below (details in Appendix A.2):
>
> |   % of labeled data     | LP (Living-17)    | PFA (Living-17)     |  LP (Entity-30)    | PFA (Entity-30) |
> | ------------ | ------------ | ------------ | ------------ | ------------ |
> | 100    | 54.7    | 72.0     | 46.4    | 65.1   |
> | 10   | 53.7    | 66.6    |  41.5   | 62.3   |
> | 1   | 49.0     | 64.6      | 46.1    | 62.1   |
> | 0.1  | 25.5     | 43.3     | 35.6    | 55.6   |
>
> Regarding why PFA could be better than linear probing (logistic regression), our main mathematical intuition and motivation to study PFA is that when the positive-pair graph has a high-rank adjacency matrix, the contrastive representations may contain many “spurious” information in various directions that is irrelevant to domain transfer. Linear probing can overuse these noisy, spurious directions and underuse the transferable direction, and thus underperform on the target domain. Adding the preconditioner effectively reduces the use of these spurious  feature directions, hence leads to theoretical guarantees on the target domain.
>
> The following experiments also corroborate our theoretical intuition that LP/logistic regression overuse nontransferable features and over-align to the decision boundary on the source .  We observe that **PFA has lower source domain test accuracy** than linear probing (LP), yet the **target domain test accuracy is higher** with PFA, as shown in the following table:
>
> |         | LP (Source)    | PFA (Source)   | LP (Target)   | PFA (Target)   |
> | -------- | ------- |-------- |--------- |-------- |
> | Living-17    | 91.3   | 90.5   | 54.7   | 72.0  |
> | Entity-30   | 84.8   | 77.3    |46.4   |65.1   |

---

> ### Comment · Reviewer_GwWK · 2022-08-09
> **Thank you for the response**
>
> I thank the authors for the response. The new experiments about effect of sample size are certainly a useful addition. Are the claims, "Linear probing can overuse these noisy, spurious directions" and "preconditioner effectively reduces the use of these spurious feature" currently just unverified hypotheses (besides the empirical finding that PFA is better)? It is not immediately clear to me why this should be the case. Any intuition/experiment/visualization that supports this will only make the claims in the paper stronger.
>
> I am of the opinion that the theoretical contributions in the paper outweigh some of the (seemingly valid) empirical deficiencies pointed out by other reviewers; better empirical results will only help the visibility of the paper. Overall I still think that passes the bar for acceptance, and I stick with my score of "Accept".

---

> > ### Author Response · Authors · 2022-08-09
> > **Intuition for the preconditioner**
> >
> > We thank the reviewer for their positive evaluation of our theoretical contribution.
> >
> > Our main intuition for the preconditioner comes from the theory. In our theory, the contrastive learned representations are essentially eigenvectors of the normalized adjacency matrix, where larger eigenvectors correspond to the clustering structure in the graph (hence can be viewed as signal), and smaller eigenvectors correspond to more nuanced intra-cluster structures in the graph (hence can be viewed as noise). The **role of the preconditioner is to keep the signal directions and weaken the noise directions**, due to the following reason: since the preconditioner is set to be the representation covariance matrix, it has larger eigenvalues in those signal directions, and smaller eigenvalues in those noise directions. As a result, after multiplying the (power of) preconditioner to the representation, those noise directions would be killed whereas the signal directions still largely remain. A more detailed discussion can be found in Section 4.3.

---

> > > ### Comment · Reviewer_GwWK · 2022-08-09
> > > **What about LP?**
> > >
> > > Thanks for the quick response. So this only argues why preconditioning on top of the "mean classifier" can be better than not using preconditioning. Does this also say why *learning* a linear classifier (using logistic regression or even just the optimal source linear classifier) might also be bad, right?

---

> > > > ### Author Response · Authors · 2022-08-09
> > > > **Analysis of LP remains an open question**
> > > >
> > > > Yes, we agree with the reviewer that our current theory only tells us why PFA works well, but doesn't tell us why the learned linear classifier (LP) would fail.
> > > >
> > > > We believe that given the existence of noise directions in the representation, it's likely that a learned linear classifier on the source would also use those noise directions (in a way that is not transferable to the target). That being said, this is indeed just a hypothesis at the current stage. We note that a rigorous comparison between learned LP vs PFA would require a tighter analysis of LP, which by itself is a challenging open question.

---

> > > > > ### Comment · Reviewer_GwWK · 2022-08-09
> > > > > **Thanks for the clarification**
> > > > >
> > > > > Worth adding this discussion in the paper for completeness about PFA.

---

> > > > > > ### Author Response · Authors · 2022-08-09
> > > > > > **Thanks for the suggestion!**
> > > > > >
> > > > > > Thank you to the reviewer for raising this insightful question, and for suggesting adding this discussion in the paper! We will include the clarification about PFA vs. linear probing in our camera ready version when given additional space.

---

### Official Review · Reviewer_1Rjf · 2022-07-13

**Rating:** 7
**Confidence:** 3
**Soundness:** 4 excellent
**Presentation:** 4 excellent
**Contribution:** 3 good

**Summary:**

The paper's goal is to understand the phenomenon of linear transferability of contrastively learnt representations. Namely, representations learnt via contrastive learning on a certain subpopulation which enable linear classification between two classes are also amenable to transfer to a classification task between the same pair of classes but on a different subpopulation to what was seen at train time.
A key underlying object which will help shed light on this phenomenon is the positive-pair graph which is a weighted graph between all possible inputs $x$. The weight of an edge between $x,x'$ is the probability of sampling $x'$ as the positive pair of $x$ or vice-versa.
Prior work which introduced the positive pair graph connects the success of contrastive learning to clustering and expansion properties of the positive pair graph. This work builds on this connection to establish expansion conditions on the positive pair graph for linear transferability to occur. The main technical criterion identified is that the connections between two example clusters belonging to the same class across different subdomains should be strictly stronger than the connections between two clusters belonging to different classes in different subdomains.
To enable linear transferability, rather than train a linear head using cross-entropy loss, the authors propose using a simple feature averaging head. This alone however can run into issues when there are complex asymmetries present in the problem. As a result, they propose a preconditioning step leading to their (Preconditioned Feature Averaging) PFA algorithm. They show that this linear head provides significantly better results empirically in transferring to out of domain settings on the BREEDS dataset.


**Questions:**

**Suggestions**
1. For the experiments can you include a comparison to the baseline contrastive approach of using the InfoNCE loss and throwing away the head of the encoder and training a linear layer on top at time of fine-tuning (as is done for instance in SimCLR)?


**Limitations:**

The authors adequately addressed the potential negative societal impact of their work.

**Strengths And Weaknesses:**

**Strengths**
1. The paper touches upon an important and poorly understood problem in self-supervised learning today and makes a novel and significant contribution in this space.
2. The writing is clear and the flow of the paper is smooth. The authors provide intuition at multiple technical places in the paper which helps the reader.
3. The paper offers a graph theoretic explanation of the reasons why certain empirical approaches in ML are successful. This is an interesting intersection between graph theory and ML.

**Weaknesses**
1. The theoretical tools and insights used seem to have a significant overlap with prior work of HaoChen et al which make the significance of the theoretical contributions of the current paper a bit diminished.

Nonetheless, I think this a technically solid paper with interesting experimental results which clears the bar for acceptance at NeurIPS.

---

> ### Author Response · Authors · 2022-08-02
> **Response to Reviewer 1Rjf**
>
> We thank Reviewer 1Rjf for the positive review, expressing that our work "makes a novel and significant contribution" on an “important and poorly understood problem”, and "clears the bar for acceptance at NeurIPS".
>
> We address the questions of the reviewers below.
>
> > "The theoretical tools and insights used seem to have a significant overlap with prior work of HaoChen et al which make the significance of the theoretical contributions of the current paper a bit diminished."
>
> We agree that we use a framework similar to [1]. However, the **goal of our analysis is different**: [1] studies how data in a cluster are mapped to similar embeddings, whereas our paper studies the relative positions of the embeddings of the clusters, which requires understanding the relative strength of inter-cluster connections.  Due to the different goal, **our theoretical analysis is novel and more challenging**.
>
> On a technical level, the analysis is more challenging than [1] also because inter-cluster connections are generally much weaker in the graph than intra-cluster connections, and thus the signal from inter-cluster connections are harder to deal with because they are more likely to be washed out by the intra-cluster connections. Essentially, we develop a technique that can take out the influence of the intra-cluster connections so that they don’t negatively influence our comparisons between inter-cluster connections.
>
> > "For the experiments can you include a comparison to the baseline contrastive approach of using the InfoNCE loss and throwing away the head of the encoder and training a linear layer on top at time of fine-tuning (as is done for instance in SimCLR)?"
>
> Thanks for the suggestion! For STL->CIFAR10 dataset, we included below the linear probing (LP) baseline approach requested by the reviewer, which is reported by [2] with the pre-mlp SimCLR features. For Living-17 and Entity-30, we ran similar experiments with a linear probe (LP) on top of pre-mlp features, but using the spectral contrastive loss (SCL) instead of InfoNCE loss due to computational constraints. (SimCLR requires a large batch size to work well.) Our understanding is that the reviewer is asking for pre-mlp features specifically, and thus we discarded the mlp encoder, and tested the linear probe accuracy. We also include two baseline UDA methods SENTRY [3] and DANN [4] for comparison. The results are below.
>
> |      | SENTRY   | DANN  |  LP (pre-mlp)   | PFA (post-mlp)     |
> | ------------ | ------------ | ------------ | ------------ | ------------ |
> | STL->CIFAR10    | 53.8   | 55.2     | 75.4    | **80.0**   |
> | Living-17    | 75.5   | 71.3     | **79.1**    | 72.0   |
> | Entity-30    | 56.1   | 57.5     | 63.8    | **65.1**   |
>
> The experiments show that pre-mlp linear probing achieves better performance compared to previous domain adaptation methods, but is still worse than (post-mlp) PFA on two out of three datasets (Entity-30 and STL->CIFAR10). Our discovery that **PFA works well with post-mlp features** may be surprising to many people, since discarding the mlp projection head has been widely accepted as a necessary yet mysterious step in various contrastive learning methods.
>
> We would like to note that the motivation of these experiments were mostly demonstrating the practical relevance of the theoretical analysis (instead of advocating a new method).
>
> References:
>
> [1] Provable guarantees for self-supervised deep learning with spectral contrastive loss. HaoChen et al. NeurIPS 2021.
>
> [2] Connect, not collapse: Explaining contrastive learning for unsupervised domain adaptation. Shen et al. ICML 2022.
>
> [3] Selective entropy optimization via committee consistency for unsupervised domain adaptation. Prabhu et al. ICCV 2021.
>
> [4] Domain-adversarial training of neural networks. Ganin et al. JMLR 2016.

---

> ### Comment · Reviewer_1Rjf · 2022-08-05
> **Thank you for the response**
>
> Thank you to the authors for their response. After briefly reading some of the other reviews with more negative scores, I feel the paper currently offers enough on the theory side that a slightly less comprehensive experimental section is not a blocker for it to be accepted. Hence, I will maintain my score.

---

### Author Response · Authors · 2022-08-09
**Overall Response**

We thank reviewers for their time and thoughtful feedback. The reviewers find our work “makes a novel and significant contribution” and “clears the bar for acceptance at NeurIPS”(1Rjf), acknowledge the “phenomenon of linear transferability is quite interesting” and “vote for accepting this paper” (GwWK), and appreciate that our paper is “well-written and easy to follow” (TUn2).

We acknowledge that reviewers TUn2 and RBh1 have concerns about experiments, and we’ve included additional experiments in our revision. We want to emphasize that our main contribution is theory, and experiments are only complementary.

(i) Main contribution: The core research contribution of this work is to provide **theoretical analyses** of why contrastive learning helps domain adaptation. As far as we know, we provide the **first** theoretical analysis that shows contrastive learning helps unsupervised domain adaptation in a general setting.

(ii) Novelty: We develop new proof techniques in our analysis, advancing the current status of contrastive learning theory. While the algorithm we study (PFA) is a version of the existing protonet method, we are the first to demonstrate (both theoretically and empirically) that this simple algorithm works well as a domain adaptation algorithm.

(iii) Role of the PFA algorithm: The PFA algorithm serves as a bridge between theory and existing empirical methods that are challenging to analyze. We study PFA because it enjoys nice theoretical properties, resembles existing methods, and is more amenable to theoretical analysis. Thus, our focus is the theoretical analysis of PFA, and we reported the empirical performance only as a supplementary component to show the practical relevance of the PFA algorithm. Somewhat surprisingly, PFA indeed achieves competitive performance for some UDA benchmark datasets, which may be of independent interest to people in a broader empirical community.

We have incorporated reviewers’ feedback, and we believe this makes our paper much stronger. Concretely, we’ve added the following results in our revision:
* Added experiments on STL->CIFAR10 dataset.
* Added comparisons with more domain adaptation baseline methods including ERM, DANN and SENTRY.
* Added comparisons with linear probing on pre-mlp contrastive features.
* Added ablation study with varying amounts of labeled source data.
* Added theoretical result (Theorem G.2) that relaxes the clustering assumption, allowing for outliers.

---

### Meta-Review · Area_Chair_AkHA · 2022-08-24

**Recommendation:** Accept
**Confidence:** Certain

**Metareview:**

The manuscript provides theoretical analyses of why contrastive learning helps domain adaptation. It developed new proof techniques, advancing the current status of contrastive learning theory. The manuscript then studied a version of the existing prototypical network approach by Snell et al., called Preconditioned Feature Averaging PFA (mean classifier + preconditioner) and demonstrated both theoretically and empirically that this simple algorithm works well as a domain adaptation algorithm.

Reviewers acknowledged several positive aspects of the manuscript including the phenomenon of linear transferability and the fact that contrastive learning helps with it. There are concerns related to experiments, however, the theoretical contributions in the manuscript outweigh some of the empirical deficiencies.

Discussion phase has addressed concerns related to more experiments and showing that the theory is robust to outliers. To further increase the impact of the manuscript in terms of theory indeed predicts several phenomena that are verified by experiments, the authors have responded positively to the suggestion about testing on much larger scale and common for SOTA unsupervised domain adaptation testing datasets such as DomainNet and VisDA. The results will be added to the camera ready version.


**Award:**

No

---

### Decision · Program_Chairs · 2022-09-14

Accept